# Evaluating 60 GHz FWA Deployments for Urban and Rural Environments in Belgium

**DOI:** 10.3390/s23031056

**Published:** 2023-01-17

**Authors:** German Castellanos, Brecht De Beelde, David Plets, Luc Martens, Wout Joseph, Margot Deruyck

**Affiliations:** 1Department of Information Technology, IMEC-Ghent University, 9052 Ghent, Belgium; 2Department of Electronics Engineering, Colombian School of Engineering, Bogota 111166, Colombia

**Keywords:** radio access network (RAN), 5G networks, Massive MIMO (MaMIMO), fixed wireless access (FWA), 60 GHz, millimetre-Waves (mmWaves)

## Abstract

Fixed wireless access (FWA) provides a solution to compete with fiber deployment while offering reduced costs by using the mmWave bands, including the unlicensed 60 GHz one. This paper evaluates the deployment of FWA networks in the 60 GHz band in realistic urban and rural environment in Belgium. We developed a network planning tool that includes novel backhaul based on the IEEE 802.11ay standard with multi-objective capabilities to maximise the user coverage, providing at least 1 Gbps of bit rate while minimising the required network infrastructure. We evaluate diverse serving node locations, called edge nodes (EN), and the impact of environmental factors such as rain and vegetation on the network design. Extensive simulation results show that defining a proper EN’s location is essential to achieve viable user coverage higher than 95%, particularly in urban scenarios where street canyons affect propagation. Rural scenarios require nearly 75 ENs per km${}^{2}$ while urban scenarios require four times (300 ENs per km${}^{2}$) this infrastructure. Finally, vegetation can reduce the coverage by 3% or increment infrastructure up to 7%, while heavy rain can reduce coverage by 5% or increment infrastructure by 15%, depending on the node deployment strategy implemented.

## 1. Introduction

Implementing 5G technologies has been challenging in the last few years. With the constant increase in the users’ bit rate demands, telecom operators require a fast and economical way to provide broadband connectivity to home and business users. Fixed wireless access (FWA) is a viable alternative compared to fiber to provide the needed connectivity with massive growth opportunities for operators due to its reduced cost compared to fiber-to-the-home (FTTH) deployments and faster time to market. It is predicted that up to 60% of geographical regions could be served by FWA, adding up to 230 million connections by 2027 [1,2].

To support more than 1 Gbps in the access network, FWA should consider wider bandwidths, typically available in mmWave frequencies. The V-band (including the 60 GHz frequencies) can deliver such bandwidths.

Since wireless link distances are reduced to a few hundred meters for the 60 GHz band, backhaul fibre connectivity is not feasible due to higher costs and long deployment times. Hence, a wireless backhaul network is a practical solution to install nodes to serve residential users in FWA networks. In these networks, the node that connects the core network (CN) with the serving nodes is the point of presence (PoP), the node that serves the residential users is called an edge node (EN), while the node located on the user’s premises is the customer premises equipment (CPE), as depicted in Figure 1. Several challenges arise from implementing such networks, such as faster and cheaper network infrastructure that meets the demand of urban and rural environments.

The following study investigates diverse aspects needed to plan an FWA network that can support 99% user connectivity for users requesting up to 1 Gbps in urban and rural scenarios. The main focus is to deliver an optimal network that accounts for realistic environments, including buildings, vegetation and rain parameters that highly affect the propagation losses. To this end, we implement an FWA network planner which includes three-dimensional (3D) ray tracing (RT) and a hierarchical backhaul topology under a multi-objective optimisation graph network based on the tool developed in [4,5,6]. The main novelties of this work are:A hierarchical FWA network planner with ray tracing capabilities to support future bit rate demands of residential users.A multi-objective optimisation algorithm that minimises the infrastructure size and power consumption while maintaining the network service quality requirements.A study of environmental parameters such as rain and vegetation that affect the propagation channel in the 60 GHz band and its impact on network performance.A survey of edge node locations for telecom operators to provide optimal network performance.

## 2. Fixed Wireless Access 

FWA systems were first thought of in the 1990s when they were utilised to replace the wiring in conventional telephone systems with the primary goal of lowering installation costs and start-up times [7]. Today they have the same purpose, except that data traffic requirements are much higher, needing larger bit rates over shorter communication distances between nodes and lower implementation costs. To support such needs, FWA systems envision the mmWave frequency bands, including the unlicensed 60 GHz band, as a solution to provide bit rates higher than 1 Gbps, thanks to recent advances in radio technology for these frequencies [8,9,10]. With carrier frequencies in the V-band (50–75 GHz) and channel bandwidths of 2.16 GHz, the IEEE 802.11ad standard defines physical and medium access control (MAC) layer interfaces for short-range, high-throughput systems, enabling SISO (single-input single-output) data rates of up to 2.5 Gbps [11]. In addition, MIMO (multiple-input multiple-output) is supported by its successor, IEEE 802.11ay [12]. This standard establishes bit rates up to 40 Gbps, and extended transmission distances up to 500 m, thanks to its channel bonding and multi-user MIMO (MuMIMO) capabilities [13]. In addition, the 3GPP Rel 17 will expand its frequency range 2 (FR2) from 52 to 71 GHz, including this band for 5G deployments [14].

A typical FWA network deployment is presented in Figure 1, where a fiber broadband city-wide network is connected to the PoP, and the PoP is connected to ENs via a wireless hierarchical backhaul network. Finally, each CPE is connected via the direct 60 GHz link to the ENs to support residential users’ connectivity.

### 2.1. Network Design

Link distances of wireless systems using carrier frequencies in the mmWave band are typically limited to 200 m, which results in significantly smaller cells than sub-6 GHz deployments. In particular, when employing directional antennas at 28 GHz in midtown Manhattan, USA, only 33.3% of receiver locations within 200 m of the transmitter experienced an outage [15]. In contrast, 64.9% of places experienced it for distances under 425 m. Similarly, at 73 GHz, it is found that only 13.9% of locations within 200 m of the transmitter experienced an outage [16]. Early studies on mmWave cell deployment show that improving the base station location diversity enhances the system’s coverage, particularly in rain and vegetation [17,18].

Authors in [19] compare an FWA deployment in the 3.5 GHz and 28 GHz bands for the urban city of Citra Raya Cikupa, Indonesia, showing that the mmWave band deployment is recommended due to lower cost, despite the fact that it requires almost doubling the nodes compared to the sub-6 GHz band. In [20], authors developed a design tool to optimise the placement of ENs and CPE working in the 28 GHz band for a non-legacy network, showing that their multi-iteration greedy algorithm outperforms traditional meta-heuristic algorithms. The optimisation of ENs placement on lamp posts in the street intersection of an urban scenario is studied in [21], showing that a 0–1 knapsack problem formulation could optimise the network deployment under certain constraints. A network-aware edge node placement problem is studied in [22]. Here, the authors developed a framework to implement an edge cloud network under 5G constraints, showing that their method can improve up to 30% the number of deployed ENs compared to heuristic and mixed integer linear programming (MILP) methods. Finally, a comprehensive tutorial on the diverse aspects of the design and implementation of 5G FWA networks is presented [23].

### 2.2. Planning Tools for FWA Designs

Network modelling and planning is a continuous process of evaluating a network to support the incremental need of users. It started in the pre-internet era when telecom operators needed to dimension their telephony networks. Now, with the evolution of technology and data services, planning a dynamic wireless network has become an enormous task. It includes the definition of the topology and architecture that models the infrastructure the closest to reality as possible. In addition, users’ traffic behaviour must be known in advance to be able to assign resources to users in the most optimal way. However, this optimal solution depends on the particular objectives of the network designer or business case that is evaluated. Additionally, based on the requirements of diverse government bodies, there is a need for network planning that accounts for all the aspects of the 5G networks, in particular, the ones that include power consumption and exposure analysis in their solutions.

Nowadays, several network planners can be used to investigate the performance of 5G networks and plan them. There are comprehensive studies that provide the latest achievements in simulation techniques and platforms for 5G networks [24,25]. However, none of them can evaluate FWA, energy consumption and the impact of exposure in the same tool. In the following subsections, the network planner tools are grouped within two focuses: (i) the industry-focused ones that aim to evaluate the performance of already deployed networks and the performance of future improvements, and (ii) the research-focused ones which aim to evaluate future technologies and methods for 5G networks. Table 1 summarises relevant 5G network planners and the aspects that they are capable to evaluate, compared with the GRAND Tool used in this paper.

#### 2.2.1. Industrial Focused Tools

Most of the industrial focused planning tools work under a licensed software model, as described in Table 1. The ASSET suit is one of the most complete 5G network planners in the market able to validate several of the 5G business models. The 5G New Radio (5G NR) modelling is provided by enhanced propagation models, complex antenna arrays, comprehensive multi-technology 3D coverage and capacity simulations, including human exposure analysis [26]. Similarly, the Atoll 5G NR module offers operators a flexible and evolving framework for designing and deploying 5G networks [27]. This is a wireless network design and optimisation software that includes multi-radio access network (RAN) modelling for 5G networks, including outdoor and mmWave propagation based on ray tracing techniques that include up to 60 GHz frequencies. It is based on Monte Carlo simulations that offer accurate planning with the capability of revising planning data based on measure campaigns. Still, it lacks FWA and hierarchical topology capabilities. Both ASSET and Atoll tools are the only ones that evaluate (measure) human exposure, but none of them optimise the network planning towards exposure. Capgemini Engineering delivers a 5G planning solution which consists of several 5G tasks to create a continuous optimisation framework for operators working in sub 6 GHz and mmWave bands aiming to support the operators’ key performance indicators (KPIs) [28].

The Cell Designer tool [29] provides network modelling and coverage analysis for 5G and other wireless technologies, using multi-vendor specific equipment modelling. On the other hand, the Hamina tool provides a cloud-based robust network planner for WiFi, 5G private networks (PN) and Bluetooth low energy (BLE). The web-based technology provides fast network designs for sub-6GHz networks with accurate 3D planning [30]. Hardware vendors, such as Huawei, provide a 5G network planner for its partner operators to solve 5G challenges such as precision propagation models for urban and rural scenarios, accurate coverage prediction, automatic site planning and planning for novel services [31].

iBwave design enterprise is a 3D network design tool for both Wi-Fi and small cell cellular private networks that can be merged with a detailed site aesthetic to perform network design. It works mainly in indoor sub-6 GHz bands to predict optimal node location placements, and FWA is not considered for its solution [32]. An other tool is the LSTelecom planning tool, which is able to map existing sites and optimises the network operation with interference assessment, providing maximum possible capacity of already deployed and upgraded 5G networks [33]. The NetSim uses the 5G NR library to simulate end-to-end across all layers of the 3GPP protocol stack. This tool is used for industry and research and can evaluate existing implementations and validate future extensions in emulated scenarios, being the only industrial tool capable of simulating aerial networks [34,55].

The Terragraph project provides an open-source Link-Level network planning tool with software defined network (SDN) capabilities for a 60 GHz mesh network backbone for an FWA use case. This software is intended to provide network planning for cloud-based network real deployments with of the shelf equipment under the open-software methodology, using either SDN or software defined radio (SDR) for low-cost implementation [1,35].

#### 2.2.2. Research Focused Tools

The range of research tools is very wide, and depends on the specific objectives the research teams investigate. They can range from channel modelling to link-level simulators. Still, a full protocol stack simulator needs optimisation functions to reduce the complexity and provide accurate results. There are about 50 5G simulator tools available in the literature, according to [56]. The following is an overview of some of the most popular tools.

The 5G Air simulator is an open-source system-level tool that models key 5G NR elements such as MaMIMO, extended multi-cast and broadcast, enhanced random access procedure, and NB-IoT, as well as support for performance analysis of reference 5G scenarios with varying mobility, traffic load, and deployment configurations [36]. Similarly, the 5G K-Simsys, an open-source software system-level simulator too, evaluates the performance of 5G networks, in particular, the new improvements of 5G NR such as beam forming in MaMIMO systems. Several antenna configurations are implemented to evaluate the mmWave performance of the 3GPP Rel 16 standard [37].

On the other hand, the MatLAB 5G toolbox is a link-level simulator that provides 5G NR standardised modelling able to simulate UL and DL base-band specifications and the effects of interference in end-to-end links. It supports 5G NR waveform generation, channel modelling, test and measurements of 5G links, including signalling and control information channels; still, network planning must be done by external tools [25,38]. CGA Simulation produced an online replica of Kensington, Liverpool, utilising ’digital twin’ technology to build a 5G network planning tool. This tool enables 5G network planning to examine an area by visualising LoS 5G receivers and working around constraints such as trees or tall buildings. It models the IEEE 802.11ad protocol working at 60 GHz to evaluate the packet error rate (PER) of the network links and compare them with test-bed measurements providing comparable results [39,40].

In [57], the authors evaluate the performance of a single 5G BS at 28 GHz for a FWA service, investigating the effects of trees on coverage for urban and suburban scenarios. It shows that CPEs installed outdoor have a range up to 160 m compared to 70 m for indoor installations. In addition, throughput could be improved if MaMIMO beamforming is used. The National Institute of Standards and Technology (NIST) has introduced a quasi-deterministic channel realisation tool that includes the IEEE 802.11ad/ay standard with ray tracing capabilities for system level simulations [58]. In [59], this tool is used to validate a simplified model of the RT methods for mobile indoor receivers in the 60 GHz band.

Discrete-event network simulators, such as OMNeT++ [42,43], OPNET [45] or NS-3 [41], are some of the most popular simulation tools. While these are essentially generic network simulators, specific modules such as 5GSim, LTESim, Simu5G are able to simulate LTE and 5G networks. These tools are packet focused and mimic substantial sections of the protocol stack due to their event-driven nature. In terms of computational complexity, this makes simulating very large networks with many network nodes prohibitively expensive. OMNeT++ includes the 4GSim, 5GSim and Simu5G modules to perform cellular network simulations focused on the radio resource allocation (RRA) procedures and data plane simulations in 4G and 5G networks. Similarly, the NS-3 network simulator with the 5G-LENA module, aims to solve real-world issues, including several 5G technologies such as hybrid beamforming, vertical handover, V2X communications and multi-tier heterogeneous network. Then, OPNET simulates the physical, MAC and Link layer of the open systems interconnection (OSI) model, featuring comprehensive hardware models that consider resource allocation and protocol testing for realistic Monte Carlo simulations.

The OpenAirInterface is a network simulator that integrates the core network with the RAN to perform complex simulations that can support 3GPP Rel 8 and other improvements such as 5G NR and OpenRAN [44]. Finally, the Vienna 5G simulator performs system-level simulations abstracting the physical and MAC layer to model the complete protocol stack in a simplistic object-oriented implementation. By doing this, the Vienna 5G simulator can simulate thousands of 5G nodes in complex scenarios, including 3D channel modelling, MaMIMO and mmWaves [46,47]. Unfortunately, none of the network planning tools described in this subsection are capable of analysing human exposure, much less planning based on electromagnetic exposure restrictions.

### 2.3. Trials

Diverse FWA trials around the world have been implemented since 2016. AT&T, in collaboration with Ericsson and Intel, deployed its first 5G FWA, providing 4K video and VoIP calls using 15 GHz and 28 GHz spectrum [60]. The Terragraph project serves 100+ homes in 1 km${}^{2}$ in Mikebuda, Hungary, providing coverage of 99.5% with top speeds of 1 Gbps for outdoor CPE and 605 Mbps for indoor ones. Nokia has achieved the highest throughput in Riyadh, Saudi Arabia in a high-density area [61]. In Romania, Orange connected 16 residential users to their 5G virtual packet core network using 26 GHz band equipment, showing data rates up to 1 Gbps for distances larger than 1 km [62]. Adtran has developed its Metnet 60G Mesh solution to extend gigabit connectivity to dense residential and business areas, using the IEEE 802.11ad standard in the 60 GHz band to provide up to 4 Gbps per node in a self-organised network (SON) to improve redundancy and service. Diverse rural communities in North America have adopted this solution to implement low-cost gigabit connectivity [63,64].

## 3. Evaluation Framework

The following section defines the network architecture and scenarios considered for our simulation analysis. We aim to evaluate the impact of the node locations and the environmental factors on the network performance.

### 3.1. Network Architecture

As described in Figure 1 the mmWave FWA network architecture is composed of three different wireless nodes:Point of presence (PoP) is the node that connects the wireless backhaul network to the fibre core. We use an PoP location from one of the operator in Belgium. It consists of a tower, 14 m in height, with four antenna boxes covering a 360-degree area.Edge nodes (EN) are the ones in charge of serving final users or create a backhaul mesh network with other edge nodes.Customer premises equipment (CPE) is the final node in users’ homes

We define a hierarchical topology, as shown in Figure 2, where CPE nodes (blue circles) are connected to only one EN (purple triangles). The backhaul network connecting ENs to the PoP has a tree shape where all traffic is aggregated to the PoP in a hierarchical way. In this architecture, each EN has only one route to connect to the PoP, which minimises the number of hops, simplifying the complexity of the ENs, thus reducing its cost [65].

### 3.2. Node Locations

Different strategies are being considered to evaluate the influence of the location of nodes on the network performance as shown in Figure 1. Here, the location of wireless nodes is defined. One hundred random houses are selected in each area of interest. The CPE has a fixed site placed on the wall closest to the street of the chosen house at 4 m of elevation or 50 cm under the roof height. A post is placed at the property’s entrance when the closest street is further away than 25 m from the building. For the last use case (see Section 3.4), only 12 buildings are selected, and the CPE elevation is 25 m.

Next, the possible ENs’ locations are placed using three strategies.

Facade only (FA): Places the ENs equipment in the same way as the CPE at 4 m on the wall closest to the street in houses, public buildings, or other constructions.Lamp post only (LP): Place the ENs on lamp posts at the height of 4 m. Since detailed information on the lamp post is unavailable, it is assumed that lamp posts are placed at corners or intersections and subsequent ones 50 m along each street until covering the investigated area.Facade and the lamp post (FP): also called joint location, it is a mixture of the previous strategies, where nodes could be on walls or lamp posts.

A comparison of the first two strategies is shown in Figure 3, where it is noticeable that for rural scenario, the lamp posts aids to connect further away nodes, while in the urban scenario, the distribution density is quite similar between facade and lamp post. In addition, it is worth mentioning that these are possible locations of the ENs, and after the allocation procedure, the algorithm selects the final used locations (See Section 4.1).

### 3.3. User Density and Traffic

We define the users’ traffic based on the operator’s inputs. The typical home subscriber’s internet consumption is 30 Mbps, while the TV consumption is 6 Mbps, based on data from the network operator. The model defines that 80% of home users use TV and Internet services (36 Mbps), and the other 20% only use the Internet (30 Mbps). In addition, a headroom allows home users to have a peak internet connection of 1 Gbps, meaning that a user could made a speed test and the network should provide this speed.

The point of presence (PoP) is defined to support a maximum of 100 user connections for an aggregated bit rate of 4.48 Gbps/PoP. This explains that the area of interest should contain only 100 connected homes. In addition, user density is determined by the density of houses in the area and the desired penetration ratio of the operator in the investigated area. These are detailed in the Use Cases subsection.

### 3.4. Use Cases

We define three use cases for the analysis of network performance:

Use Case 1 (UC1) is focused on supporting last-mile connectivity in rural scenarios of already connected areas. For Belgium, the number of rural houses per km${}^{2}$ is 150. It is assumed that the operator has a penetration of 50% of the homes, so the total density of served houses is 75 per km${}^{2}$. The rural town of Leest in Flanders is selected with an area of 1.33 km${}^{2}$ to account for the 100 CPE served.

Use Case 2 (UC2) is focused on connecting new urban cities with a home density of 1100 houses per km${}^{2}$. The city centre of Liege is selected, with an area of approximately 0.45 km${}^{2}$ and operator penetration set to 20%. Similar to UC1, UC2 also serves 100 CPEs.

Use Case 3 (UC3) studies the impact on the beam elevation service for taller buildings and larger aggregated traffic. Specifically, in each tall building, the CPE provides internet service for eight apartments with an internal cable connection between them and the CPE. This implies that the CPE must serve an aggregated traffic of 250 Mbps for the internet plus 48 Mbps for television and headroom of 1 Gbps. A tall building is defined as an apartment construction with an altitude larger than 50 m. An urban residential area in Ghent is selected, where up to 12 residential towers, equivalent to 96 apartments, are found. Table 2 summarises the use cases, and Figure 4 depicts the areas.

A final evaluation of the performance of the losses due to reflection is performance. Note that up to now UC1 to UC3 only considers direct link on the 60 GHz band. To evaluate if improvement on the performance of the calculation, a multi-ray bouncing analysis is performed to UC1 and UC2 where up to two reflected rays are accounted for the total path losses as described in Equation (Equation 3).

### 3.5. Link Budget and Channel Modelling

The considered system operates in the V-Band, particularly in channels 1–6 for the IEEE 802.11ad standard [12], with a bandwidth of 2.16 GHz per channel. One channel is used for the access network (CPE-EN), while up to four channels are considered for the backhaul network, depending on capacity requirements. Table 3 lists the parameters utilised by the transceiver to calculate the link budget parameters in the network. Detailed realistic 3D antenna gains and beamforming codes implemented are modelled based on specific hardware design [66]. In addition, Table 4 defines the required signal-to-noise ratios (SNR) used to achieve the requested bit rates for the spirit module from confidential information provided by Pharrowtech partner [66]. All CPE nodes should support at least 1 Gbps (SNR = 3.5 dB), while the backhaul links at least 2.5 Gbps (SNR = 7.5 dB). Higher modulation schemes are used to improve link usage.

The 60 GHz channel model is based on actual measurements in this band [67]. The path loss attenuation is modelled as follows:(1)$$PL\left(d\right)\left[\mathrm{dB}\right]=PL_{p}+PL_{\mathrm{refl}}+PL_{\mathrm{rain}}+PL_{\mathrm{veg}}$$
where $PL_{p}$ is the loss due to propagation, $PL_{\mathrm{refl}}$ is the reflection loss, $PL_{\mathrm{rain}}$ is the loss due to the rain and $PL_{\mathrm{veg}}$ the excess loss due to vegetation.
(2)$$PL_{p}\left[\mathrm{dB}\right]=PL_{0}+10\eta \log_{10}\left(d\right)+\chi$$

For $PL_{p}$, $PL_{0}$ is the reference path loss at 1 m ($PL_{0}$ = 71 dB), η is the the path loss exponent (η = 1.78) and χ is a lognormal variable representing shadowing fading with σ = 3.5 dB.

$PL_{\mathrm{refl}}$ is the attenuation due to the reflection of the multi-bouncing ray tracing. We implement a two bouncing ray tracing method, where each ray launches a ray every 5 degrees, looking for an intersection in a circumference with a 10 m radius. Up to two wall bounces are accounted until a ray is discarded [68]. We select two types of walls: bricks and concrete, representing the material of the reflecting wall. We assume that concrete is for new houses with a probability of appearance of 30%, while 70% of the houses are considered to be made of brick [6]. The reflection loss depends on the bouncing angle α and the type of material and is described in Equation (Equation 3).
(3)$$PL_{\mathrm{refl}}\left[\mathrm{dB}\right]=\left\{\begin{array}{ccc} 15\left(1-\frac{\alpha }{90}\right) & \mathrm{Brick} & 70\%\\ 8\left(1-\frac{\alpha }{90}\right) & \mathrm{Concrete} & 30\% \end{array}\right.$$

The $PL_{\mathrm{rain}}$ is the attenuation due to rain, and it depends on the area where it is used. For Belgium, a 0.1% outage probability is found for a rain intensity of 10 mm/h, which is considered light rain. For heavy rain, a 0.001% outage probability corresponds to 50 mm/h [69]. The rain attenuation measurements we presented in [70] for light rain are consistent with the values reported in the ITU-P 530-16 838 [71,72,73]. In addition, the heavy rain values tend to differ in different regions; the attenuation value for western Europe is reported to be slightly higher than in other areas, particularly in fixed wireless links [74]. We select the values in Equation (Equation 4) for our simulation since they are consistent with our measurements shown in [70].
(4)$$PL_{\mathrm{rain}}\left[\mathrm{dB}/\mathrm{km}\right]=\left\{\begin{array}{cc} 5 & \mathrm{Light}\;\mathrm{rain}\\ 20 & \mathrm{Heavy}\;\mathrm{rain}\; \end{array}\right.$$

Attenuation due to vegetation $PL_{\mathrm{veg}}$ can be modelled using the exponential decay model which has the form of $PL_{\mathrm{veg}}=A*f^{B}*d^{C}$. Three important models describe this attenuation at mmWaves, Weissberger [75], Cost 235 [76] and FITU-R Vegetation [77]. Vegetation loss measurements at 60 GHz presented in [67], confirm that the Weissberg is a better fit for larger distances, but for typical urban/suburban tree densities up to 9 m wide, the FITU-R model is a better fit. We select the FITU-R model for winter (without leaves) and summer (with leaves), as shown in Equation (Equation 5).
(5)$$PL_{\mathrm{veg}}\left[\mathrm{dB}\right]=\left\{\begin{array}{cc} 0.37*f^{0.18}*d^{0.59} & \mathrm{Winter}\\ 0.39*f^{0.39}*d^{0.25} & \mathrm{Summer} \end{array}\right.$$

## 4. Network Planning and Optimisation

### 4.1. Simulation Tool

To execute the FWA mmWave study, we designed a simulation tool which performs network planning simulations based on realistic 3D scenarios, accounting for multiple optimisation functions to find the most appropriate network topology [4,5,6,51].

The main novelties of the tool used in this study include where multi-bouncing ray tracing up to second-order reflections is needed and meshing capabilities for the backhaul network in the 60 GHz band. The new algorithm is presented in Figure 5.

*STEP 1*: In the initialisation, the configuration files are loaded, and the scenarios defined. Those inputs include the following files:Shapefile of the environment with the area information, such as buildings, vegetation and streets needed to determine the location of nodes.The bit rate information of users, as defined by the requirements of the simulations.The technology files describing the information needed to perform the link budget calculation as shown in Section 3.5.Other configuration files such as the type of node location and type of environmental restrictions (rain or vegetation).*STEP 2*: The possible EN location generation defines the possible location of the nodes in the environment. It first determines the type of location required on the facades, lamp posts or both. Then, it assigns many possible sites based on a maximum number of ENs and evaluates if those are viable. This evaluation is based on Dijkstra’s shortest path algorithm [78], where all the nodes must have at least one path to the PoP. The connections of each path are calculated based on the path loss between the nodes and minimum supported bit rate (i.e., 2.5 Gbps). Those possible EN locations that do not have a connection to the PoP are removed from the list. This process is called tree pruning and is shown in Figure 2. Then, the CPEs are located on the facades of randomly selected houses, and the bit rate and specific location are assigned to the CPE node.*STEP 3*: Once the location of all the nodes is known, a graph network generation is constructed for both CPEs and ENs nodes. First, a list of possible links from each CPE to all close ENs is created. For each connection, a link budget is calculated based on the requested bit rate and the path loss between nodes. The path loss calculation includes ray tracing and environmental factors such as rain and vegetation. A link is added to a viable link list if it fulfils the network requirements. Similarly, the backhaul network between ENs is created, creating links between ENs and accounting for an aggregated minimum bit rate of 2.5 Gbps for each link.*STEP 4*: The constraints and requirements for the graph network are assigned in this step. First, it identifies the optimisation objectives described in Section 4.3. The rules of each node and link are described in a model that is read by the mixer-integer programming (MIP). In this case, the problem objective is to maximise the number of connected users, while minimising the cost of the network which includes the infrastructure size, the accumulated path loss and the averaged hop count.*STEP 5*: The MIP solver uses the Gurobi Optimiser to solve the multi-optimisation problem based on linear expression to map the objectives of the problem into a possible solution [79]. Once one or several solutions are found, the best one is assigned to the wireless network. In particular, once all the radio resources are allocated, each CPE node is assigned to an EN, and the backhaul network is created, calculating the capacity of each link and aggregating all the traffic up to the PoP node.*STEP 6*: In the last step, the algorithm calculates the network performance metrics, saving them in files for further post-processing. In addition, MATLAB scripts were created to collect and aggregate the results and print them, as shown in the results section.

### 4.2. Ray Tracing Algorithm

To improve the accuracy of the links between the nodes, a multi-bouncing ray tracing method is implemented. The following algorithm resides in creating possible links in Step 3 of Figure 5. The proposed algorithm is divided into three sub-steps and presented in Figure 6, where a set of rays launched, and interception nodes are presented.

Ray Launching: Each ray is launched every θ degrees starting for the azimuth of the transmitter antenna and rotating either clockwise or anti-clockwise depending on the location of the receiver node. In these simulations, the value of θ is 10 degree.Ray tracing: For each ray, two reflections are accounted. Each ray is a construction of consecutive points every meter until the maximum distance is reached, maximum PL is achieved, and the maximum number of rebounds are counted or until a receiver node is reached. For bouncing rays, each reflection is calculated based on the angle of arrival of the ray and the angle of the bouncing wall.Ray interception: As each ray is composed of points, each point evaluated is a receiver node within a distance of $r=d*\tan(\frac{\theta }{2})$. If a node is within that radius, a possible link is created for the pair of Tx-RX. Instead, if after 250 points (250 m) an Rx is not intercepted, the next ray is evaluated until the whole antenna aperture is scanned.

### 4.3. Multi-Objective Methods

As multiple objectives must be simultaneously satisfied during the simulations, a linear utility function to solve the multi-objective problem is created. It finds the possible solutions with compromises that satisfy the restriction of the network operation. The problem is defined as follows:

First, each objective variable is described, and then the multi-objective minimisation function is presented.

The *user coverage* variable $f_{Cov}$ is important, with a weight of 40%. It aims to maximise the number of CPE connected to the core network. Equation (Equation 6), describes the user coverage where $CPE_{ij}$ is a boolean value to identify the connection between $CPE_{i}$ to $EN_{j}$, normalised by the total number of CPEs in the area.
(6)$$f_{Cov}=1-\left(\sum_{i=0}^{{Total}_{CPE}}\sum_{j=0}^{{Total}_{EN}}\frac{CPE_{ij}}{Total_{CPE}}\right)$$

The *backhaul utilisation* variable $f_{BU}$, has a similar importance to the user coverage since it intends to minimise the mesh network infrastructure, providing the requested user coverage. Equation (Equation 7) presents this function where $EN_{j}$ is a boolean function for an active EN *j*, normalised by the total number of possible ENs in the network.
(7)$$f_{BU}=\sum_{j=0}^{Total_{EN}}\frac{EN_{j}}{Total_{EN}}$$

The *hop count* variable $f_{HC}$ is a crucial function that aims to reduce the delay of the network by reducing the number of hops that a connected CPE needs to reach the PoP. The $HopCount_{j}$, in Equation (Equation 8), represents the total number of hops that the $EN_{j}$ needs to reach the PoP, normalised by the maximum number of hops $EN_{j}$ can use to reach the PoP.
(8)$$f_{HC}=\sum_{j=0}^{Total_{EN}}\frac{HopCount_{j}}{Max_{hopcount}}$$

The *path Loss* variable $f_{PL}$ aims to minimise the aggregated PL of a specific connection with the final objective of increasing the available supported bit rate of the link. It calculates the path loss $PL_{ij}$ between the $CPE_{i}$ and the $EN_{j}$, and the aggregated path loss of all the *k* links associated with that specific path towards the PoP as shown in Equation (Equation 9).
(9)$$f_{PL}=\sum_{i=0}^{Total_{CPE}}\sum_{j=0}^{Total_{EN}}\left(PL_{ij}+\sum_{k=0}^{Total_{links}}PL_{jk}\right)$$

Finally, Equation (Equation 10) describes the *power consumption* variable $f_{PC}$ that minimises the whole network power consumption by calculating the power used by each active $EN_{j}$, as shown in [4], normalised by the maximum power consumption of all the available ENs. This final function has the lowest importance, with a weight value of only 5%.
(10)$$f_{PC}=\sum_{j=0}^{Total_{EN}}\frac{PowerConsumption_{j}}{Max_{PowerConsumption}}$$

The multi-objective minimisation function written in canonical from is described in Equation (Equation 11), where the objective variables are $f_{Cov}$, $f_{BU}$, $f_{HC}$, $f_{PL}$ and $f_{PC}$.
(11)$$\begin{array}{cc} \min_{}:\; & w_{1}f_{Cov}+w_{2}f_{BU}+w_{3}f_{HC}+w_{4}f_{PL}+w_{5}f_{PC}\\ s.t.:\; & C_{1}:\forall i,j\in N;0\leq CPE_{ij}\leq 1\\ \; & C_{2}:\forall i,j\in N;0\leq \sum RB_{ij}\leq EN_{RB}\\ \; & C_{3}:\forall i,j\in N;\sum EN_{ij}\geq 1,\forall i\neq j\\ \; & C_{4}:\forall i\in N;\sum EN_{0i}\geq 1\\ \; & C_{5}:\forall i,j,k\in N;PL_{ij}\leq PL_{max};PL_{jk}\leq PL_{max}\\ \; & C_{6}:w_{1}+w_{2}+w_{3}+w_{4}+w_{5}=1 \end{array}$$

The values of $w_{1}$ to $w_{5}$ represent the weights of the independent objective variables. $w_{1}$ = $w_{2}$ = 40% are the weights for the user coverage and backhaul usage objective functions since they are the most important objectives in this problem. $w_{3}$ = 10% is the weight of the hop count objective variable. $w_{4}$ = $w_{5}$ = 5% are the values for the path loss and power consumption variables. The values of *i*,*j* and *k*, represent the *i*th CPE, the *j*th EN and *k*th PoP nodes, respectively.

The multi-objective minimisation function is also subject to several network constraints. $C_{1}$ defines that a CPE can be connected to only one EN. $C_{2}$ specifies that the number of resource blocks (RB) used per EN must not exceed the maximum capacity of 10,800 RBs or its equivalent maximum bit rate. $C_{3}$ specifies that all ENs in the networks need a possible, stable connection with at least another EN. $C_{4}$ indicates that the PoP needs at least one connection with some ENs. $C_{5}$ restricts the path loss of the links to be used only if they are smaller than the maximum supportable path loss for the defined minimum bit rate, which is 1 Gbps for the CPE, 2.5 Gbps for the ENs and 4 Gbps for the PoP. Lastly, $C_{6}$ defines the normalisation of the weights in the minimisation function.

### 4.4. Methodology

A network planning algorithm is implemented to study the proposed architecture. We investigate the statistical stability of the simulations by running a series of simulations until specific simulation metrics (Section 4.5) complete a deviation of +−0.5% of the average value of the metric over 100 simulations. As a result, we found stability for most metrics with 25 simulations. This leads to running 25 independent simulations for each configuration to obtain an accurate value of the investigated metric.

As defined previously, for UC1 and UC2, we investigate the impact of environmental factors such as rain and vegetation and the location of ENs. For these use cases, we compare no rain (NR), light rain (LR) and heavy rain (HR) and no vegetation (NV), winter, with vegetation (VE), summer. For the location of ENs, we compare the facade (FA), lamp post (LP) and both facade and lamp post (FL) techniques. This adds up to 18 simulation configurations for each use case. For the UC3, only the location of ENs is investigated, assuming the best-case scenario for the urban case where neither rain nor vegetation is accounted for. Finally, the multi-bouncing ray tracing technique for a line-of-sight (LoS) and non-line-of-sight (NLoS), up to two reflecting orders, is investigated.

### 4.5. Study Metrics

In order to evaluate the network solution, we define the following simulation metrics.

*Coverage*, refers to the percentage of served CPEs compared to all the CPEs in the network.*Serving Distance*, determines the usable distance between node either CPE-EN or between ENs.*Backhaul Capacity* is divided in two parts, the *supported capacity*, referring to the maximum bit rate the links in the backhaul network could provide, while the *served capacity* is the actual throughput used in those links.*Link Count* establishes the number of hops needed for a CPE to connect to the PoP.

## 5. Simulation Results

### 5.1. UC1: Rural Environment

Figure 7 shows an example of the rural solution in summer without rain. In the figure, 100 CPEs (blue circles) are served with only one PoP (green square) and 116 ENs (red triangles). The served CPE links have an average service distance of 35 m with a maximum of 372 m. The longest backhaul has nine hops, carrying an aggregated traffic of 1.126 Gbps for the last link connected to the PoP. The following simulation results from the UC1 describe the impact of environmental factors on the network performance.

#### 5.1.1. Rural Coverage

Figure 8 compares the impact of environmental and EN location on user coverage and network utilisation, and shows that the user coverage of the system depends on the EN locations. Neither FA (93.6%) nor LP scenario (95.5%) could support the required 99% user coverage.

The rain impacts the user coverage, reducing it by 2% for light rain and 5% for heavy rain. Fortunately, for FL, the user coverage can be maintained above the required limit of 99%, at the expense of the required ENs. Thus, their usage increments by 6% in light rain and 15% in heavy rain. The vegetation (summer) reduces the user coverage by 3% compared to no leaves (winter). The worst case scenario, heavy rain in summer, requires 20% more ENs/km${}^{2}$ (108 EN/km${}^{2}$) than the best case with only 89 EN/km${}^{2}$, being able to serve up to 75 homes/km${}^{2}$, leading to a ratio of 1.4 ENs per served CPE.

#### 5.1.2. Rural Serving Distance

Figure 9 shows the distribution of the serving distances for the CPE-EN and between EN–EN. The serving distance of each link helps to determine how the system is optimised. The $p_{90}$ serving distance between CPE and ENs for FA scenario is 130 m while 140 m for LP, which is reduced for FL down to 70 m. This is due to the optimisation function to maximise user coverage and reduce path loss, reducing service distance.

In the case of the distance between EN–EN, the distribution behaves differently due to the minimisation function of the network infrastructure. Here, 90% of ENs are connected within 100 m. In addition, the heavy rain reduced the connection distance by 18% and 8% in light rain. Similarly, vegetation reduces the $p_{90}$ distance by 10% in the backhaul network and only 5% in the access network.

#### 5.1.3. Rural Backhaul Capacity

The ENs’ serving capacity determines how efficiently the resource allocation process is. In our simulations, 100% of the links support more than 1 Gbps, and their maximum supported bit rate depends on the EN locations. This means that the $p_{90}$ supported bit rate is 3.6 Gbps and 3.2 Gbps for FA and LP. The $p_{90}$ for the joint design (FL) is 2.8 Gbps due to the optimisation function that maximises the user coverage, increasing the required ENs and reducing the supported bit rate of each. This is further evidenced by the fact that each EN’s provided capacity is only 500 Mbps for FL, but these values are 950 Mbps and 800 Mbps for FA and LP scenarios, respectively. Since more than 50% of links use less than 10% of the supported capacity, link efficiency is also relatively low. Comparatively, the FL scenario’s $p_{90}$ efficiency is 35% compared to 42% for the FA and 62% for the LP, demonstrating that the system’s efficiency is sacrificed in order to meet the coverage criteria.

#### 5.1.4. Rural Link Count

The link count determines the number of hops a CPE must cross until reaching the PoP. Here, the lower number of hops corresponds to FL with an average of 9 hops, which is consistent with a higher number of required ENs, and a lower averaged bit rate as seen previously. In addition, the optimisation function $f_{HC}$ aids in reducing this value in an environment where several EN are available, such as for FL. In contrast, FA location uses an average of 12 and for the lamp post scenario, 14 hops are needed since a reduction is impossible without reducing the user coverage.

### 5.2. UC2: Urban Environment

The urban environment solution in Figure 10 illustrates UC2. As observed, the backhaul connectivity shows a street canyon shape, which clearly describes an urban environment as opposed to the rural instance of UC1.

#### 5.2.1. Urban Coverage

Figure 11 shows that the use of lamp posts in an urban environments affects the coverage. Only scenarios that use lamp posts (LP and FL) can cover more than 97% of users, while the facade only (FA) can cover up to 50% of users. This is because nodes on walls need a greater beam angle, resulting in lower antenna gains due to the street canyon structure of urban cities.

The lamp post cases have additional beam angle flexibility, providing the tool more freedom to establish a transmitter with the highest gain. Since the coverage is only lowered by 0.5%, vegetation has little effect in urban settings. This is mainly because there are only a few bushes and tiny trees in the downtown districts, which barely affect the performance of the network as a whole. Instead, torrential rainfall causes the coverage to decrease by 1% while boosting EN consumption by 5%. With a 3% difference between the worst case (382 $\mathrm{EN}/{\mathrm{km}}^{2}$) and the best case (370 $\mathrm{EN}/{\mathrm{km}}^{2}$), the environmental factor generally has a negligibly small impact.

#### 5.2.2. Urban Serving Distance

When employing the joint (FL) scenario, the $p_{90}$ servicing distance is significantly decreased (40 m as opposed to 100 m in other instances), as there is a higher chance of serving an EN located nearby. This is also due to the optimisation method that aims to minimise the path loss to maximise the supported bit rate. Vegetation only reduces the $p_{90}$ serving CPE distance by 3%, while heavy rain reduces it by 8%. Oppositely, the $p_{90}$ backhaul connecting distance increases by 6% for the FL scenario due to the optimisation algorithm that aims to maximise network size and minimise hop count. Here, rain significantly impacts the distance since it shortens the backhaul-connected distance by 20% and 6% during a heavy and moderate rainfall, respectively.

#### 5.2.3. Urban Backhaul Capacity

Because of the shorter distances mentioned above, the backhaul capacity is 25% larger than in a rural setting. The $p_{90}$ supported capacity surpasses the 4 Gbps for LP only and 4.4 Gbps for FL. However, for FA, LP, and FL, the actually $p_{90}$ served capacity is only 1 Gbps, 530 Mbps, and 300 Mbps, respectively. The average number of users served by each EN decreases due to the increased number of serving ENs and the lower likelihood of LoS between nodes. As a result, the FL link efficiency is just 20%.

#### 5.2.4. Urban Link Count

The $p_{90}$ link count for both lamp post cases is nine hops. It is similar to the rural scenario, showing that the hop count does not depend on the environment. The hop count is only slightly affected by vegetation, rising to 5%, while the rain raises it by 10%.

### 5.3. UC3: Tall Buildings

Use Case 3 is a particular case for the urban scenario, where CPE nodes are in the middle of taller buildings in urban areas. Simulation results show that, similar to the urban scenario, the lamp post usage is essential to achieve 100% user coverage. If only the facade is used, the coverage is reduced to 35% due to the limited number of buildings in the investigated area, as shown in Figure 12.

The backhaul connection distances are similar to the urban and rural scenarios, confirming that the backhaul distances are independent of the EN location or the nodes’ averaged bit rate since all cases are required to allocate the same bit rate. However, the serving distance is comparable with the rural scenario ($p_{90}$ is 110 m) since the optimisation method aims to reduce the number of hops and the required bit rate per CPE (250 Mbps) is still under the minimum requested bit rate of 1 Gbps. As a result, the supported bit rate is only 2 Gbps showing a link efficiency of nearly 20%.

Finally, the most notable difference is the average node utilisation of only 28 ENs/km${}^{2}$ which is much lower than the 375 of the UC2. However, this is because the number of CPE is eight times lower, leading to a similar ratio of ENs needed per CPE, which is 3.7 for UC2 and 3.5 for UC3.

### 5.4. Use Case Network Comparison

This subsection compares the three use cases. User coverage in all cases can be satisfied if lamp posts are used, as shown in Figure 13. Here, more than 90% coverage for the rural and more than 97% for urban environments is achieved only for scenarios where lamp posts are used. This is due to the lamp post’s flexibility, providing better LoS and closer proximity to the CPE nodes. In contrast, the facade can only connect less than 40% of homes in urban areas, whereas it can connect more than 90% of homes in rural areas. This is because households are more dispersed, and more useful links can be established in rural settings.

In urban environments, the usage of ENs/km${}^{2}$ is four times higher than in rural cases. This is mostly caused by the three times higher population density and greater difficulty locating suitable links due to the street canyon shape, as opposed to rural locations. Additionally, this decreases the ratio of CPE served by active ENs, which is 0.8 for urban areas and 1.0 for rural areas.

The backhaul connectivity distances are similar for all scenarios, independent of the EN location. This is because the optimisation function maximises the user coverage, thus the cover area size, while maintaining the users’ required bit rate and reducing the path loss. As a result, the average connected distance is driven by the minimum supported bit rate of 2.5 Gbps maintained in all cases and is set to 100 m. However, environmental factors affect the backhaul connectivity distances more in rural environments. Rain, for example, reduces the backhaul connectivity distance by 18% in rural and 8% in urban. For the vegetation, the reduction is 10% in rural and 3% in urban environments as there is less vegetation present in urban environments.

The service capacity per EN in an urban environment (300–500 Mbps) is two-thirds compared to the rural enviroment, mainly due to the increase in the ENs. Oppositely, the supported capacity is higher in the urban scenario (4 Gbps), primarily due to the reduced serving CPE distance that supports higher bit rates.

### 5.5. Multi-Ray Bouncing Analysis

To investigate the impact of the ray tracing (RT), including multiple bouncing rays in the 60 GHz band, UC1 and UC2 are simulated again in comparison to the single ray tracing. Results reveal that even though RT is a detailed and precise simulation approach, there is only a little incremental change in the simulation metrics with a high simulation time cost, as indicated below.

For the rural scenario (UC1), only 1.5% of possible links use a reflecting ray. All of the used links are direct LoS beams. This means that none of the reflected links has sufficiently lower path loss to be selected as a serving link. Due to this, the user coverage and the number of ENs are similar.

For the urban scenario (UC2), around 3% of possible links are NLoS, and nearly 2% of the used links are NLoS. This is due to the street canyon characteristic of the downtown urban environment. The results have a small increment in the user coverage from 96.6% to 98.0% (1.4% increment). However, the number of used ENs increments by more than 5%.

In addition, the improvement depends on the EN locations. For FA, no difference was found. However, for LP, the user coverage is slightly higher (98.3%) compared to FL (98.0%), mostly due to the optimisation of the number of ENs used.

Finally, simulation time was the most affected metric. FA lasted 7.1 h per simulation, while LP 16.7 h and FL nearly 37 h. This is an increment of more than 50 times the simulation time without the multi-ray reflection. In conclusion, the single ray analysis is sufficient to account for the needed infrastructure and the requested coverage, since the reflected rays had little impact.

## 6. Conclusions and Future Work

In this study, we analysed the network performance of FWA networks working on the 60 GHz band in urban and rural scenarios. Results show that a user coverage above 95% is possible if lamp posts are included as a possible node location, in particular, in an urban scenario where a street canyon environment is predominant. Regarding the network infrastructure, the urban scenario requires four times more infrastructure than rural scenarios, but it is highly dependent on the CPE density with an average of 1 EN per CPE. In addition, the supported capacity that could reach up to 4 Gbps, four times the experienced serving capacity per EN.

In case of heavy rain, the network requires about 15% more infrastructure for the rural scenario and about 5% in the urban scenario to maintain similar user coverage. In addition, vegetation has a limited impact on coverage, but it increases the required infrastructure by 4% in the rural scenario and 1% in the urban.

This study demonstrated that FWA in the 60 GHz band is viable if an optimised backhaul network planning is utilised. To achieve this, a good knowledge of the environment is needed to account for all the constraints that a real deployment in the mmWave requires.

In the future, the location of the CPE in the house remains an open subject of interest for operators since it could determine if a user can buy off-the-shelf equipment and install it indoors or if more specialised equipment is needed to be installed on roofs or terraces. The validation of the most optimal weights for the proposed problem aiming to solve different trends for FWA deployments need to be evaluated in future simulations. In addition, multi-frequency channel use in the mmWave bands is necessary to overcome the 60 GHz bands’ downsides and provide 100% user coverage. Other areas of future research include the implementation of intelligent reflecting surfaces (IRS), mesh topology and other network optimisation techniques such as load balancing for multi-PoP networks.

## Figures and Tables

**Figure 1 sensors-23-01056-f001:**
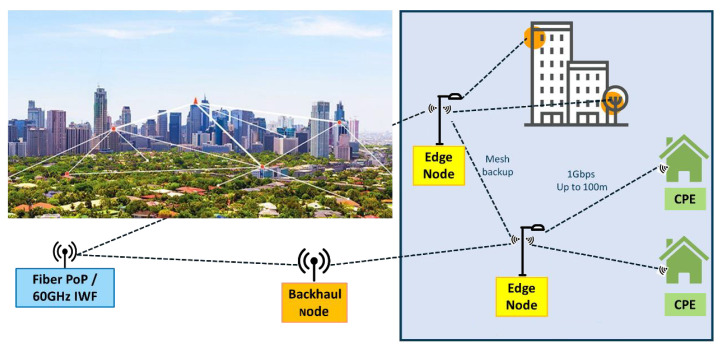
Fixed wireless access network topology in the 60 GHz Band. (PoP = point-of-presence, IWF = inter-working function, CPE = customer premises equipment) [3].

**Figure 2 sensors-23-01056-f002:**
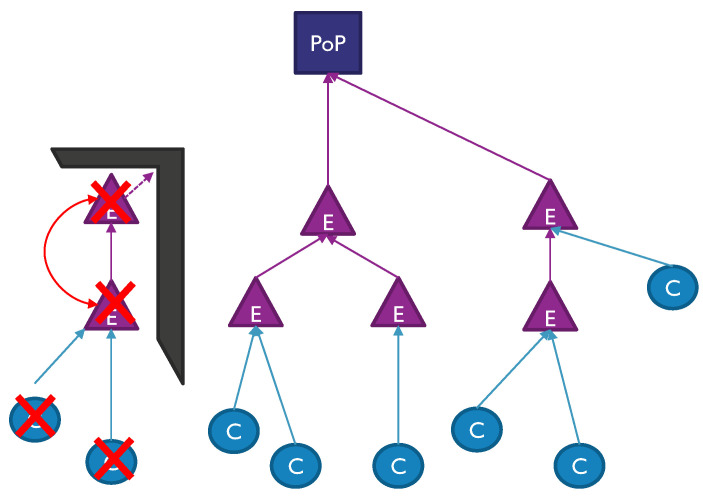
General tree network architecture with a example of pruning method (Section 4.1).

**Figure 3 sensors-23-01056-f003:**
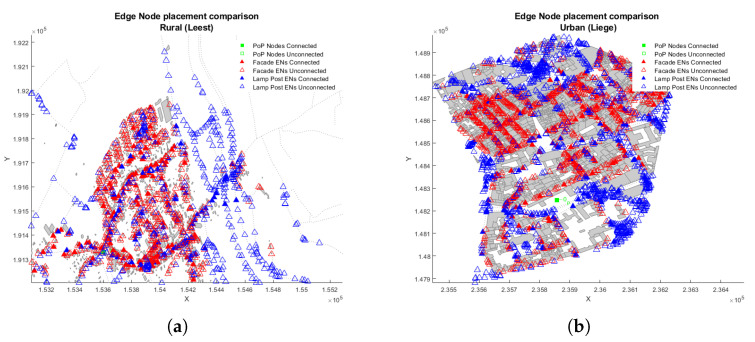
Edge node placement comparison. (**a**) Leest-Rural. (**b**) Liege-Urban.

**Figure 4 sensors-23-01056-f004:**
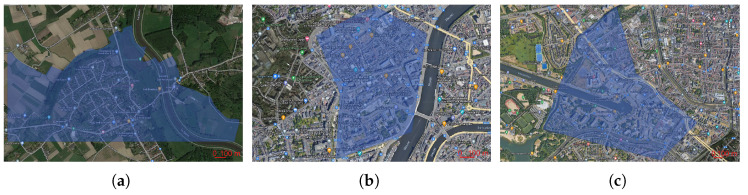
Use cases simulated areas of interest. (**a**) Leest—Rural. (**b**) Liege—Urban. (**c**) Ghent—Tall Buildings.

**Figure 5 sensors-23-01056-f005:**
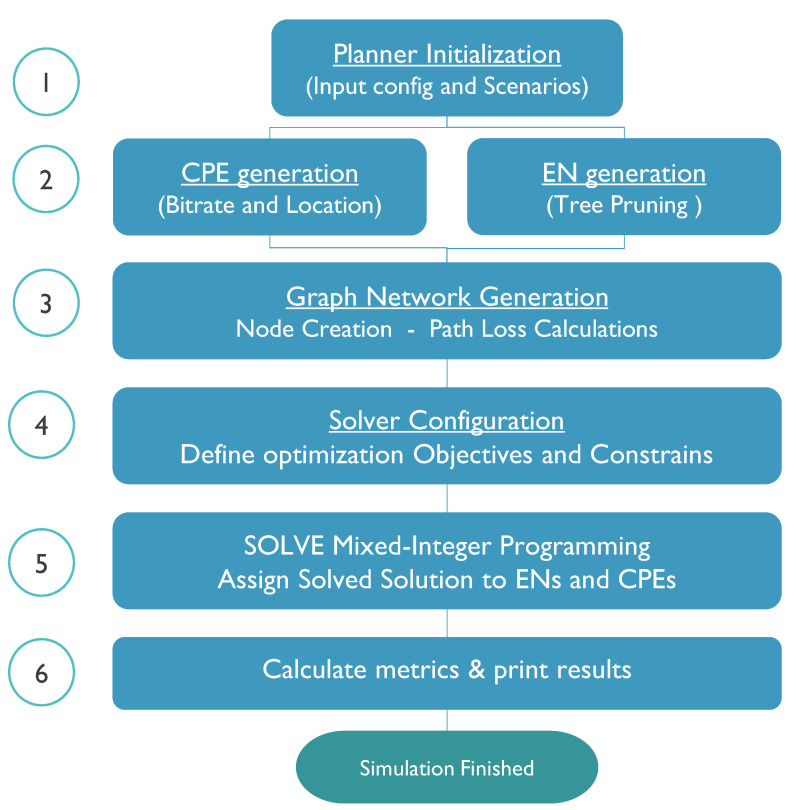
General network planning algorithm.

**Figure 6 sensors-23-01056-f006:**
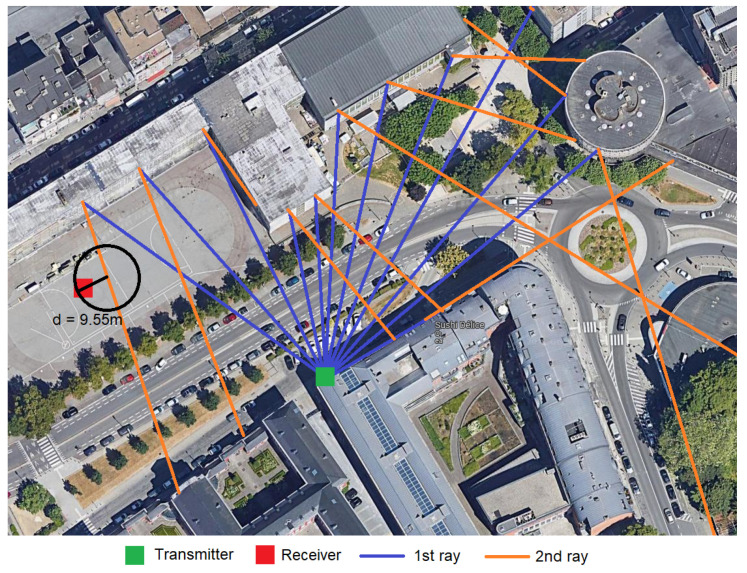
Ray tracing algorithm example.

**Figure 7 sensors-23-01056-f007:**
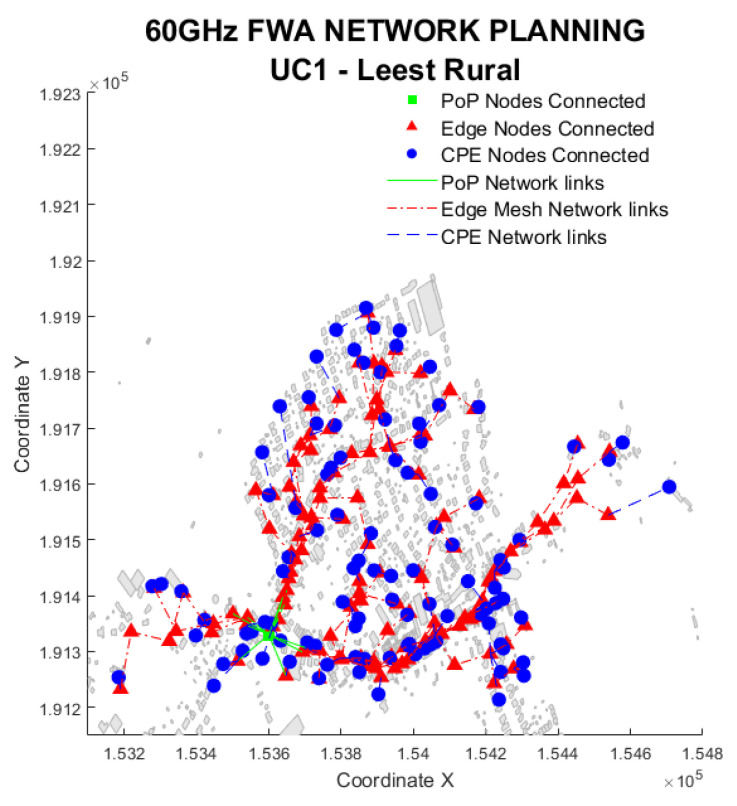
Use Case 1 solution example. Snapshot of the rural Leest area.

**Figure 8 sensors-23-01056-f008:**
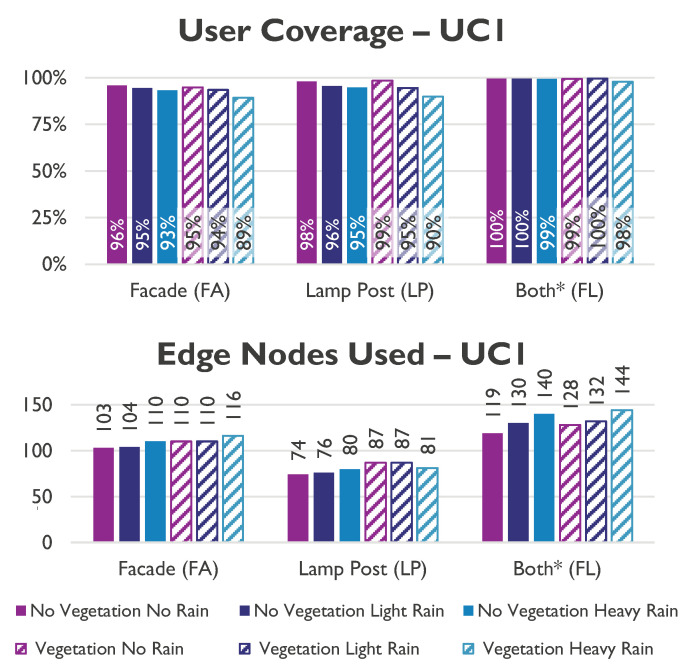
User coverage and network utilisation in UC 1.

**Figure 9 sensors-23-01056-f009:**
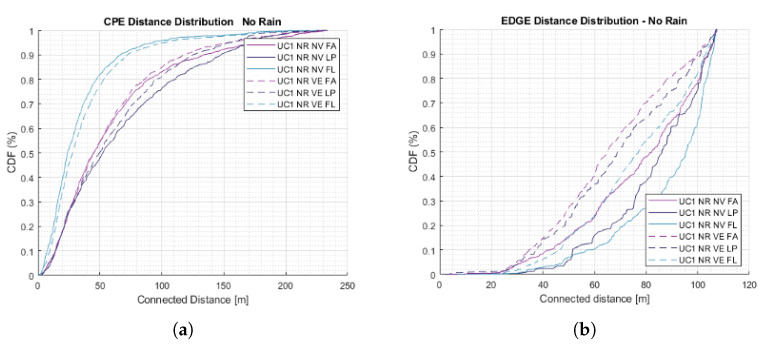
Distance distribution. (**a**) Between CPE and ENs. (**b**) Between ENs.

**Figure 10 sensors-23-01056-f010:**
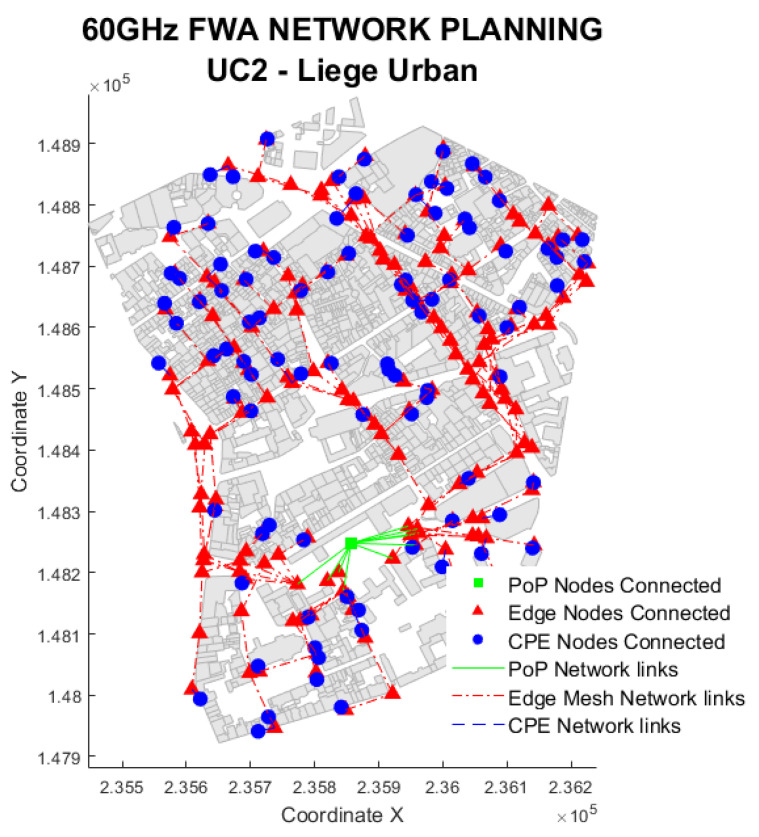
Use Case 2 solution example. Snapshot of the urban Liege area.

**Figure 11 sensors-23-01056-f011:**
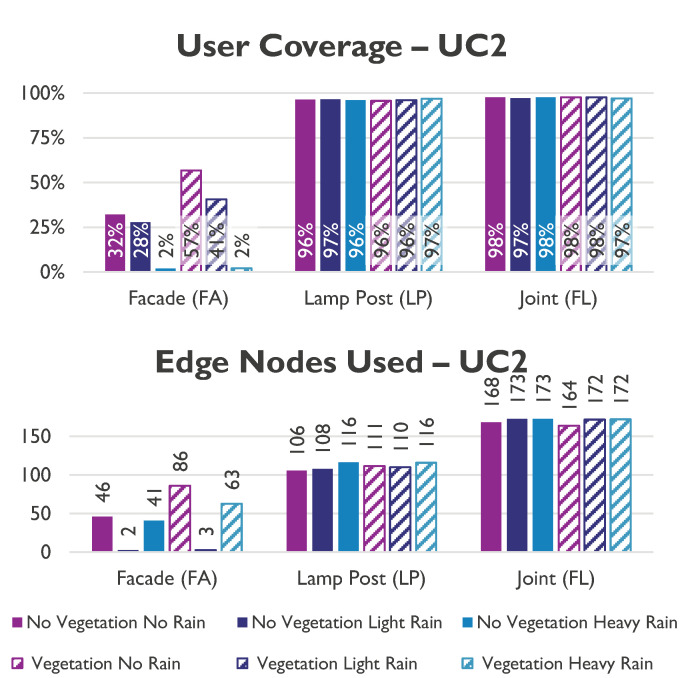
User coverage and network utilisation in UC 2.

**Figure 12 sensors-23-01056-f012:**
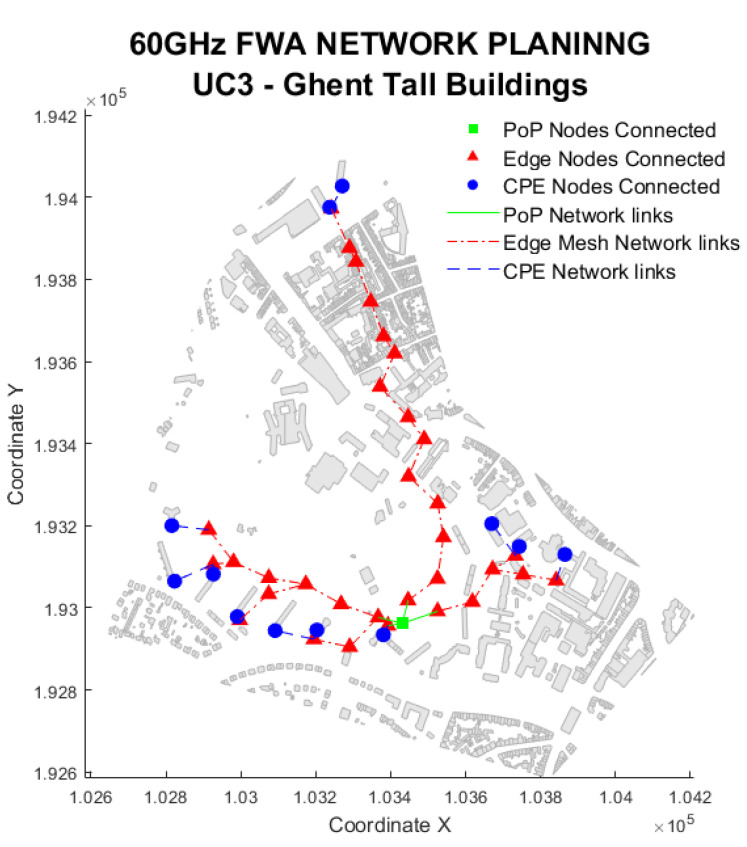
Use Case 3 solution example. Snapshot of the urban Ghent area.

**Figure 13 sensors-23-01056-f013:**
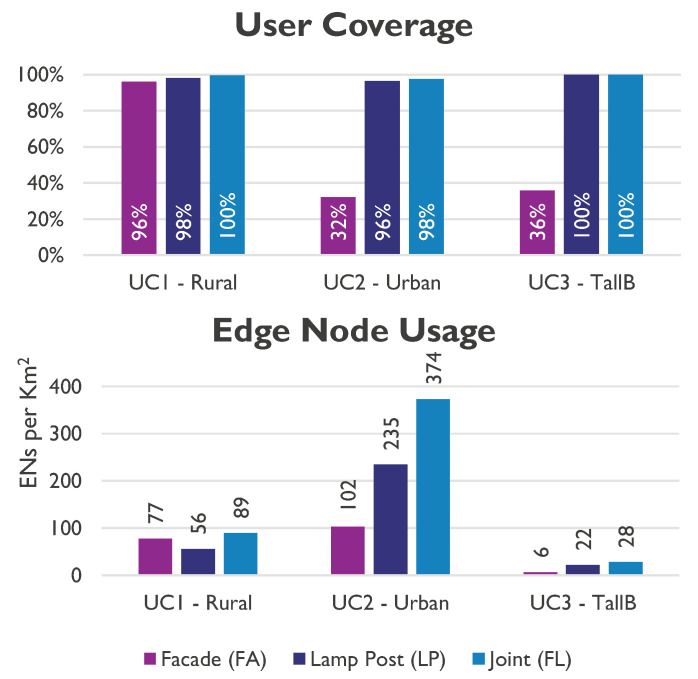
Use case comparison of coverage and network utilisation.

**Table 1 sensors-23-01056-t001:** Most relevant tools for 5G network modelling and planning.

Network Planner	Main	Software	Code	Solving	Technologies			5G			Ref
	Focus	Type	Language	Method	mmWaves	Cellular	FWA	MaMIMO	RT-GIS	Backhaul	
ASSET	Industry	Licensed	-	-	Yes	Yes	Yes	Yes	Yes	Yes	[26]
Atoll	Industry	Licensed	-	-	Yes	Yes	-	Yes	Yes	Yes	[27]
Capgemini	Industry	Service Based	-	-	Yes	Yes	-	Yes	-	Yes	[28]
Cell Designer	Industry	Licensed	-	Stochastic/MC	-	Yes	-	Yes	Yes	Yes	[29]
Hamina	Industry	Licensed	-	-	-	PN	-	-	-	-	[30]
Huawei	Industry	Licensed	-	-	Yes	Yes	-	Yes	Yes	Yes	[31]
iBwave	Industry	Licensed	-	-	Yes	Yes	-	Yes	Yes	-	[32]
LS Telecom	Industry	Licensed	-	-	Yes	Yes	-	Yes	-	Yes	[33]
NetSim	Industry	Licensed	C, R	Discrete Event	Yes	Yes	-	Yes	-	Yes	[34]
Terragraph	Industry	Open Source	Open/R	-	Yes	Yes	Yes	Yes	-	Yes	[1,35]
5G Air Simulator	Research	Open Source	C++	Event-driven	-	Yes	-	Yes	-	-	[36]
5G K-Simsys	Research	Open Source	C++	Time-driven	Yes	Yes	-	Yes	-	-	[37]
5G Toolbox	Research	Licensed	MatLAB	-	Yes	Yes	-	Yes	Yes	-	[25,38]
CGA	Research	Licensed	-	-	Yes	Yes	Yes	-	-	-	[39,40]
NS-3 (5G-LENA)	Research	Open Source	C++, Phy	Event-driven	Yes	Yes	-	Yes	-	-	[41]
OMNet++	Research	Open Source	C++	Event-driven	-	Yes	-	-	Yes	-	[42,43]
OpenAirInterface	Research	Open Source	C++	-	-	Yes	-	Yes	-	-	[44]
OPNET	Research	Licensed	C++	Stochastic/MC	Yes	Yes	-	Yes	-	Yes	[45]
Viena 5G	Research	Open Source	MatLAB	Event-driven	Yes	Yes	-	Yes	-	-	[46,47]
GRAND Tool	Research	Proprietary	Java	Heuristic/MILP	Yes	Yes	Yes	Yes	Yes	Yes	[6,48,49,50,51,52,53,54]

**Table 2 sensors-23-01056-t002:** Use case description.

	Name	City	Environment	Operator Penetration	Size [km${}^{2}$]	CPE Total [CPE]	CPE Density [CPE/km${}^{2}$]
UC1	Last Mile in Flanders	Leest	Rural (150 h/km${}^{2}$)	50%	1.33	100	75
UC2	Connected in Wallonia	Liege	Urban (1100 h/km${}^{2}$)	20%	0.45	100	220
UC3	Tall Buildings	Ghent	Urban (1100 h/km${}^{2}$)	50%	1.5	12 (96 homes)	8 (64 homes)

**Table 3 sensors-23-01056-t003:** Transceiver network parameters for the simulation.

Parameter	Unit	Value
Frequency Band	GHz	60 (57–71)
Channel bandwidth	GHz	2.16
TDD duty cycle DL	%	75
TDD duty cycle UL	%	25
Spatial duty cycle	%	15
TX RF Paths	#	32
TX Patches/Path	#	2
TX Patch Gain	dBi	5
TX Subarray Gain	dBi	2
TX Total antenna Gain	dBi	25.1
Total radiated power	dBm	39
TX array feed loss	dB	2.5
Receiver noise figure	dB	10.2
Implementation Margin	dB	4
Azimuth Steering	°	+−45
Elevation Steering	°	+−30
Beam Codebook size	#	1024

**Table 4 sensors-23-01056-t004:** SNR values for the spirit module.

MCS	Constellation	Backoff	Datarate	SNR
		[dB]	[Mbps]	[dB]
1	BPSK	0	385	−1.4
2	BPSK	0	770	0.5
3	BPSK	0	960	2.0
4	BPSK	0	1150	3.5
5	BPSK	0	1250	4.2
6	BPSK	0	1350	5.5
7	QPSK	2	1540	3.5
8	QPSK	2	1930	4.8
9	QPSK	2	2310	6.5
10	QPSK	2	2500	7.5
11	QPSK	2	2700	8.5
12	16QAM	4	3080	9.5
13	16QAM	4	3850	11.5
14	16QAM	4	4620	12.8
15	16QAM	4	5000	13.8
16	16QAM	4	5390	15.5
17	64QAM	6	4620	14.5
18	64QAM	6	5780	16.5
19	64QAM	6	6390	18.5
20	64QAM	6	7500	19.5
21	64QAM	6	8080	21.0

## Data Availability

Not applicable.

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
