# Peer review of "Evaluating 60 GHz FWA Deployments for Urban and Rural Environments in Belgium"

_sensors, 2023, doi:10.3390/s23031056_

Round 1

Reviewer 1 Report

The authors design a simulation tool to optimize the 5G network planning considering different use cases with different channel modelling. The paper is well-written and the design is well-supported by the results.

Comment 1: In Figure 4, step 3, is this step only applied to the user's side? step 4, should be described in detail what is the objective of the problem.

Comment 2: In the multi-object minimization function equation (11), what is the optimization variables? what is the objective of the MILP? what are the given variables?

Comment 3: Why is the value of the weight factor is selected as 40% (w1, w2), 10% (w3), and 5% (w4,w5)? Can these values be selected at a different value, for example, 20%, 30%, etc. There is also no constraint for weigh factors w1 to w5 in MILP (11), how to define the value of the weight factor?

 Comment 4: there are some typos, please check. 

Reviewer 2 Report

This work presents a comprehensive evaluation of 60GHz FWA in three cases. It can give valuable information for the mmW FWA network planning to other researchers.

The followings are minor revision points.

1.     Line 11: sq km --> km^2

2.     Line 11: four times infrastructure --> not clear what it means.

3.     Line 61: larger than --> higher than

4.     Line 99: Duplicate use of ‘planning’ looks not good

5.     Line 124, 140: the comma may not be needed.

6.     Line 259: where --> if or when may be better

7.     Line 266-267: Does it mean that lamp posts are installed much denser than ‘FA’? Please make it clear how the installment of the ‘FA’ and ‘LP’ are different. You may use an additional figure or diagram to explain this.

8.     Table 2: ‘km^2’ in the items need to be deleted.

9.     Line 315: Is the reference [66] open to public? Readers must be able to retrieve the detailed information for [66] and ‘Spirit Module’ if they want.

Round 2

Reviewer 1 Report

The authors already answer to all questions and I have no questions .